# The Impact of $BiF_3$ Doping on the $Yb^{3+}$ to $Yb^{2+}$ Reduction during the $LiYF_4$:$Yb^{3+}$ Crystal-Growth Process

Amir Khadiev [1], Niyaz Akhmetov [1], Stella Korableva [1], Oleg Morozov [1,2], Alexey Nizamutdinov [1,*], Vadim Semashko [1,2], Maksim Pudovkin [1] and Marat Gafurov [1,*]

1 Institute of Physics, Kazan Federal University, Kremlevskaya, 18, 420008 Kazan, Russia
2 Zavoisky Physical-Technical Institute, FRC Kazan Scientific Center of RAS, Sibirskii Ave., 10/7, 420029 Kazan, Russia
* Correspondence: anizamutdinov@mail.ru (A.N.); mgafurov@gmail.com (M.G.)

**Abstract:** Here, we report on the opportunity to suppress $Yb^{3+}$ to $Yb^{2+}$ dopant-ion reduction in $LiYb_xY_{1-x}F_4$ mixed crystals during growth processes, using the Bridgmen–Stocbarger technique in graphite crucibles in vacuum. This was carried out by the additional doping of the $LiF$-$YF_3$-$YbF_3$ powder mixture with 1% of $BiF_3$ additive. The crystals of $LiY_{0.8}Yb_{0.2}F_4$ and $LiY_{0.8}Yb_{0.2}F_4$ with $BiF_3$ doping in the charge, were grown. The spatial distribution of the spectral-kinetic properties of $Yb^{3+}$ and $Yb^{2+}$ ions along the grown crystalline-boules were studied. It was established that the $Yb^{2+}$ concentration rises during the $LiY_{0.8}Yb_{0.2}F_4$ crystal-growth processes without the $BiF_3$ additive: the absorption coefficient of $Yb^{2+}$ ($\pi$-polarization) at 340 nm rises from 0 (at the beginning of the boule) to 2.5 cm$^{-1}$ (at the end of the boule). In contrast, the undetectable absorption of $Yb^{2+}$ along the crystals grown from the $BiF_3$ doped melt was displayed. The luminescence-decay time of $Yb^{3+}$ decreases from 3.7 to 1.8 ms from the beginning to the end of the $LiY_{0.8}Yb_{0.2}F_4$ boule grown from the $BiF_3$ undoped melt, and stays constant (~3.7 ms) along the samples grown with $BiF_3$. Here we demonstrate a positive effect of $BiF_3$ doping on the optical homogeneity of $LiYF_4$:$Yb^{3+}$ crystals.

**Keywords:** fluoride crystals; impurity ion-distribution; impurity ion redox-processes; rare-earth ions; $Yb^{2+}$ and $Yb^{3+}$ ions spectroscopy

## 1. Introduction

$LiYF_4$ fluoride crystals doped with $Yb^{3+}$ ions are well-known continuous-wave laser materials with room temperature tuning-ranges of 997–1065 nm and 998–1076 nm for $\sigma$ and $\pi$ polarizations, respectively [1]. Due to the large gain-bandwidth, the $LiYF_4$:$Yb^{3+}$ crystals can be implemented as an active medium to obtain laser oscillation in a subpicosecond-pulse regime under cryogenic cooling [2]. The unique energy-level structure of $Yb^{3+}$ ions determines the high efficiency of the optical-refrigeration effect. For example, an active medium based on $LiYF_4$:$Yb^{3+}$ can reduce the temperature the of GaAs/InGaAs double-heterostructure with 2 μm thickness, down to 165 K [3].

Common problems which reduce efficiency for both laser performance [4–6] and laser cooling [7] is the conversion of Yb ions from a trivalent to a divalent state. Ions such as $Sm^{3+}$ and $Eu^{3+}$ also demonstrate the same tendency. The optical properties of $Yb^{2+}$ ions were investigated in oxides [8], various fluorides [9,10], and fibers [5]. Due to the completed 4f electron shell of $Yb^{2+}$, its absorption spectrum is formed by interconfigurational 4fn→4fn-1 5d transitions, corresponding to the 170–400 nm spectral range [11]. In addition, the intervalence charge transfer can be observed in $Yb^{3+}$-$Yb^{2+}$ mixed systems [12], which may affect the luminescent properties of $Yb^{3+}$.

Part of the $Yb^{3+}$ ions distributed in the host-matrix can be reduced to divalent state under specific crystal-growth conditions [13]. In the work [14], the authors identify some of these conditions, namely: (a) an absence of oxidizer, (b) the trivalent rare-earth ion



must substitute the cation with the different valence, (c) a proximity of ionic radii of the trivalent activator and the divalent host-cation, and (d) an appropriate host-compound conducive to dopant valence-reduction. Generally, the $LiYF_4$ host-matrix does not satisfy any of these conditions and does not have sites for the divalent dopant; therefore, the divalent-ytterbium appearance should be associated with impurities or defects. However, the most urgent problem for the optically perfect fluoride-crystal-growth process is the presence of incontrollable oxides and the $OH^-$ group containing impurities in the charge. Indeed, the water traces always exist in the growth-chamber environment [15].

A common way to deal with $Yb^{3+}$-$Yb^{2+}$ conversion is the conduction of crystal-growth procedures under a fluorinating atmosphere, for example, in the presence of HF or $CF_4$ gases [16]. As mentioned above, despite the effectiveness of this method, the use of a fluorinating atmosphere is related to notable technical difficulties. It is worth keeping in mind the fact that the additional Bi-doping procedure can improve the optical properties of rare-earth-doped aluminosilicate, borate, phosphate and germanate glasses [17]. This occurs due to the high electron-affinity of $Bi^{3+}$ ions and the consequent effective reduction into a lower valence-state [18]. In addition, it is known that the $Yb^{3+}$ and $Bi^+$ ions being doped to the matrix as a pair mutually stabilizes their valence, which was shown in the $PbF_2$ matrix, where the conditions are in favor of $Yb^{2+}$ ions formation [19]. In this study, we demonstrate the fact that the addition of $BiF_3$ to the $LiF$-$YF_3$-$YbF_3$ charge reduces the divalent-ytterbium-formation efficiency in $LiYF_4$:$Yb^{3+}$-solid solutions grown by Bridgeman technique in graphite crucibles in vacuum.

## 2. Materials and Methods

### 2.1. Material Synthesis

The crystals of $LiY_{0.8}Yb_{0.2}F_4$ were grown in graphite crucibles from the melt with and without 1% of $BiF_3$ additive, using the Bridgman technique. The fluorides of the Yttrium, Ytterbium and Bismuth powder-mixture was used as the charge. During the growth procedure, the pressure inside the crystal-growth chamber was maintained at $10^{-4}$ mbar. The melt was pulled out from the high temperature zone through a region of a thermal gradient ($\approx 100$ °C/cm). To obtain the specific optical *c*-axis-orientation (perpendicular to the cylinder of the boule), a seeding crystal was placed inside the crucible.

Samples were cut into a parallelepipeds with $5 \times 5 \times 30$ mm typical dimensions. The optical axis of the crystals was perpendicular to one of the carefully polished facets.

### 2.2. Spectral-Kinetic Characterization of the Samples

The absorption spectra of the samples were obtained via the one-beam method, using StellarNet SL5 Deuterium/Halogen Light Source (spectral range 190–2500 nm, StellarNet, Inc. 14390 Carlson Circle Tampa, FL, USA). The experimental set-up is presented in Figure 1.

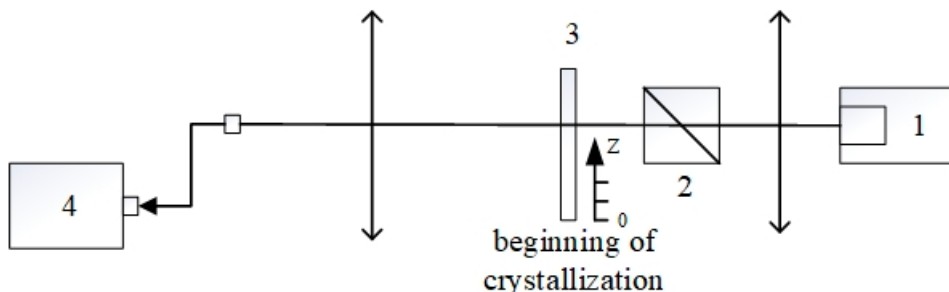

**Figure 1.** The experimental setup for registration of the absorption spectrum of the samples. (1) broad-band StellarNet halogen and deuterium light-source; (2) Glan–Taylor prism; (3) the sample; (4) CCD StellarNet spectrometer. The samples were moved along the *z*-axis that is perpendicular to the probe beam.

A Glan prism was used as the polarizer. Incident light (the probe beam) was focused on the sample surface using a fused-silica lens with a 5 cm focal length. Luminescence and absorption spectra were registered using the StellarNet CCD spectrometer (optical resolution up to 0.5 nm, StellarNet, Inc. 14390 Carlson Circle Tampa, FL, USA). The relative accuracy of the light-intensity measurements was approximately 0.5%. Luminescence excitation was carried out using pulse-periodic radiation of the InAlGaAs laser diode ATC-C1000-100 (FWHM 3 nm, "Semiconductor devices" LLC, Moscow, Russia) operating at 935 nm wavelength. Excitation laser-pulse duration and pulse repetition rate were 5 ms and 10 Hz, respectively. Luminescence-decay curves were detected using the monochromator DMR-4 (OPTICS LLS, Moscow, Russia), the photomultiplier FEU-62 (ZAPADPRIBOR LLC, Moscow, Russia, 400–1200 nm operating spectral range), and the Bordo 211 digital oscillograph (AURIS LLC, Moscow, 10 bit and 200 MHz bandwidth).

## 3. Results and Discussion

*Absorption and Luminescence Spectrum of $LiY_{0.8}Yb_{0.2}F_4$ and $LiY_{0.8}Yb_{0.2}F_4:BiF_3$ (1%)*

Spin-orbit interaction splits the $^2F$ term of the $Yb^{3+}$ ion into two $^2F_{5/2}$ and $^2F_{7/2}$ manifolds, where $^2F_{7/2}$ is the ground state. These manifolds split into three and four Stark sub-levels, respectively. The polarized absorption spectra of the crystals are shown in Figure 2.

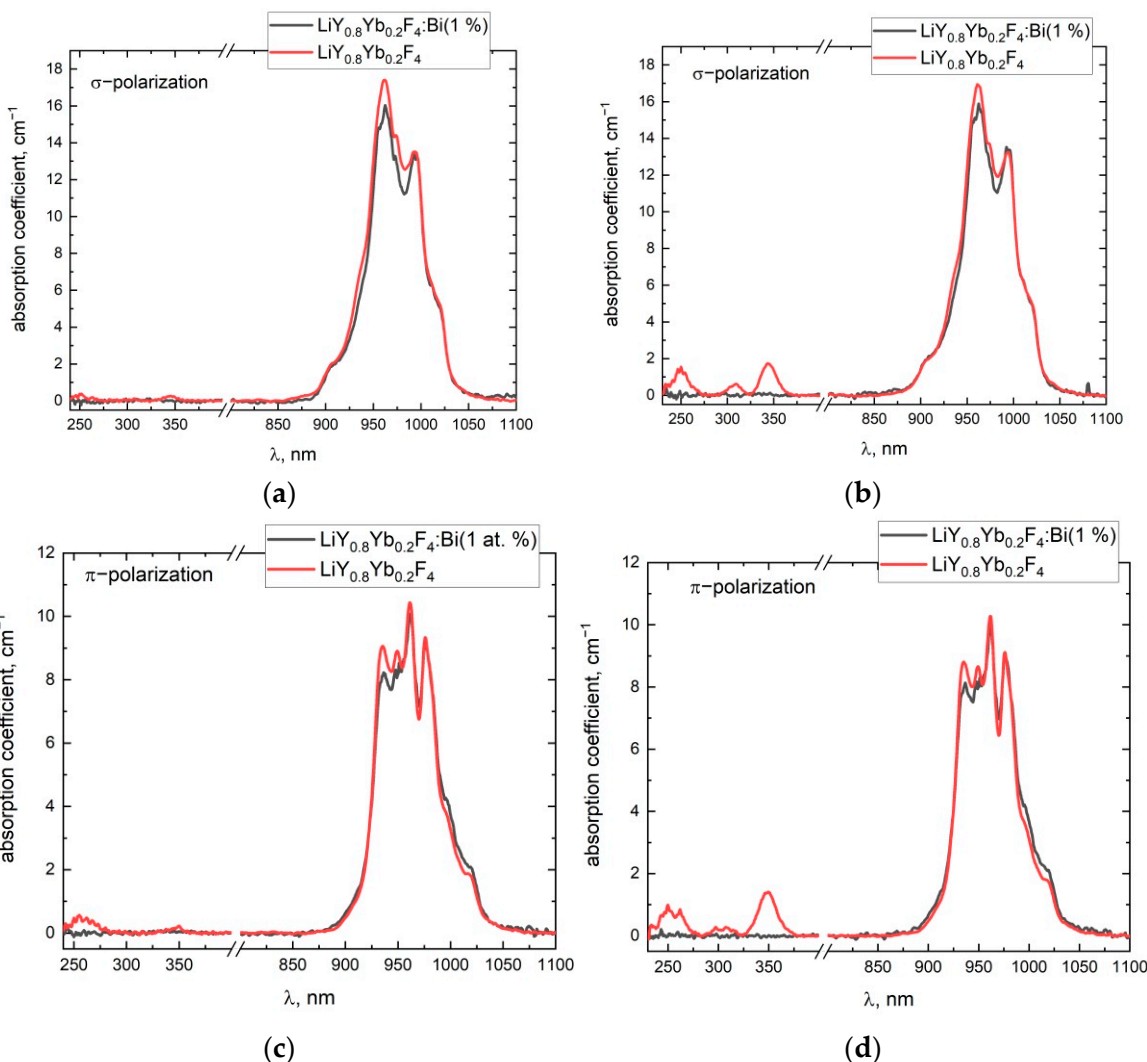

**Figure 2.** Polarized absorption spectra of two different parts of the $LiY_{0.8}Yb_{0.2}F_4$ and $LiY_{0.8}Yb_{0.2}F_4:BiF_3$ (1%) crystals (**a**,**c**)—the beginning of crystallization, (**b**,**d**)—the end of crystallization.

Each infra-red spectrum represents six absorption bands, due to the selection rules for each polarization [20]. These results are in agreement with an already published paper [21]. Four absorption peaks in the ultraviolet spectral-range can be distinguished at 248 nm, 262 nm, 307 nm, and 347 nm. These absorption bands are also in good agreement with the M. Kaczmarek et al. paper [22], in which the $Yb^{2+}$ impurity centers in $LiYF_4$ crystals were induced by X-ray radiation. In our study, $Yb^{2+}$ centers can appear in $LiYF_4:Yb^{3+}$ crystals during the growth process because the vacuum and the contact of the melt with the graphite crucible are the reducing conditions. In addition, the absorption spectra change significantly in different parts of the crystal boule. It can be seen from Figure 2, that the absorption bands in the UV range are more intense for the part corresponding to the end of the crystallization process, whereas the bands are almost not pronounced at the beginning of crystallization. The nonpolarized normalized-luminescence-spectra of the $LiY_{0.8}Yb_{0.2}$ and $LiY_{0.8}Yb_{0.2}$ with $BiF_3$ additives (1 at. % in the charge) samples under 934 nm laser-excitation, are represented in Figure 3.

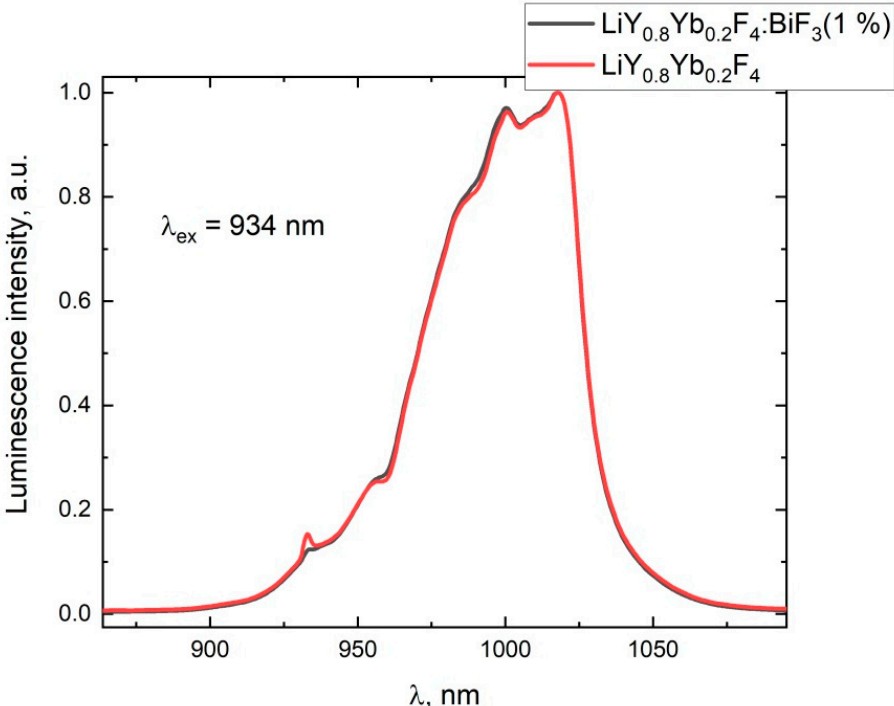

**Figure 3.** The normalized luminescence-spectra of the $LiY_{0.8}Yb_{0.2}$ and $LiY_{0.8}Yb_{0.2}:BiF_3$ (1%) samples under 934 nm laser-excitation.

The $BiF_3$ additive does not change the luminescence spectra of the $Yb^{3+}$ ions in the samples, but affects the fluorescence-kinetic properties. The drastic shortening of the luminescence-decay time of the $Yb^{3+}$ ions along the crystalline boule (with growth time) is observed for the samples without the $BiF_3$ additive in the charge (Figure 4). In contrast, the $Yb^{3+}$ life-time was constant for any part of the sample grown from the $BiF_3$-doped charge.

Due to the simultaneous increase in absorption in the ultraviolet spectral-range and the shortening of the $Yb^{3+}$ luminescence-decay time, it can be assumed that there are energy-transfer processes between $Yb^{2+}$ and $Yb^{3+}$ ions in $LiYF_4:Yb^{3+}$ single crystals. This can be associated with the well-known fact that $Yb^{3+}$ and $Yb^{2+}$ ions can form complex charge-transfer states [23]. The origin and the real mechanism of the interaction require further investigation.

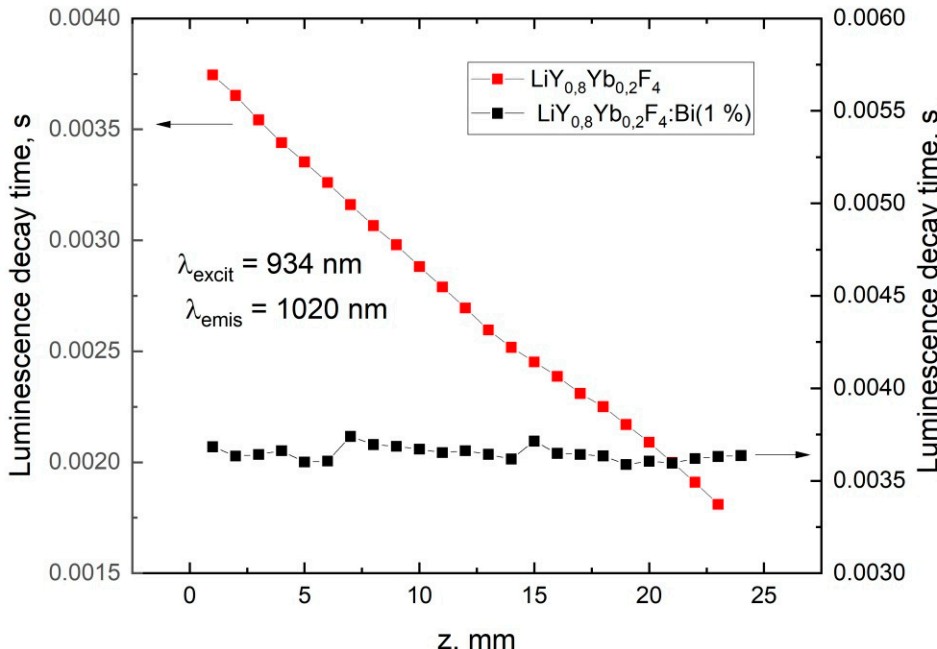

**Figure 4.** Luminescence-decay time of Yb$^{3+}$ (1020 nm) under 934 nm excitation along the crystals (along *z*-axis).

It is important to note that the presence of Bi ions in the samples does not manifest itself in the optical spectra. There are two reasons for this: (1) the bismuth in any valence state is absent or the concentration is too low in the samples, due to the volatility of the Bi-components; (2) the Bi has a valence state in the grown crystal which cannot be detected in the visible and near-IR spectral-ranges. That is why we have studied the impact of the BiF$_3$ doping effect on the spectroscopic properties of Yb$^{3+}$ and Yb$^{2+}$ ions along the crystalline boule.

The dopant concentration is proportional to the absorption coefficient, and their distribution can be described by the Gulliver–Pfann law [24]:

$$C(g(z)) = C_0 * k_{eff} * (1 - g(z))^{k_{eff}-1},\tag{1}$$

where $k_{eff}$ is the effective dopant-segregation coefficient, *g(z)* identifies the specific part of the crystal which can be expressed as: *g(z) = z/L*, where *L* is the total length of the sample and *z* is the distance from the beginning of crystallization.

Figure 5 shows the dependencies of the polarized absorption-coefficient at 340 nm of the LiY$_{0.8}$Yb$_{0.2}$F$_4$ crystal samples grown from the melt, with and without 1%of BiF$_3$ additive along the direction of growth, where z = 0 corresponds to the beginning of sample crystallization. The sample of crystal grown from the BiF$_3$-doped melt demonstrates no detectable absorption in the ultraviolet spectral-range at any part of the crystal boule.

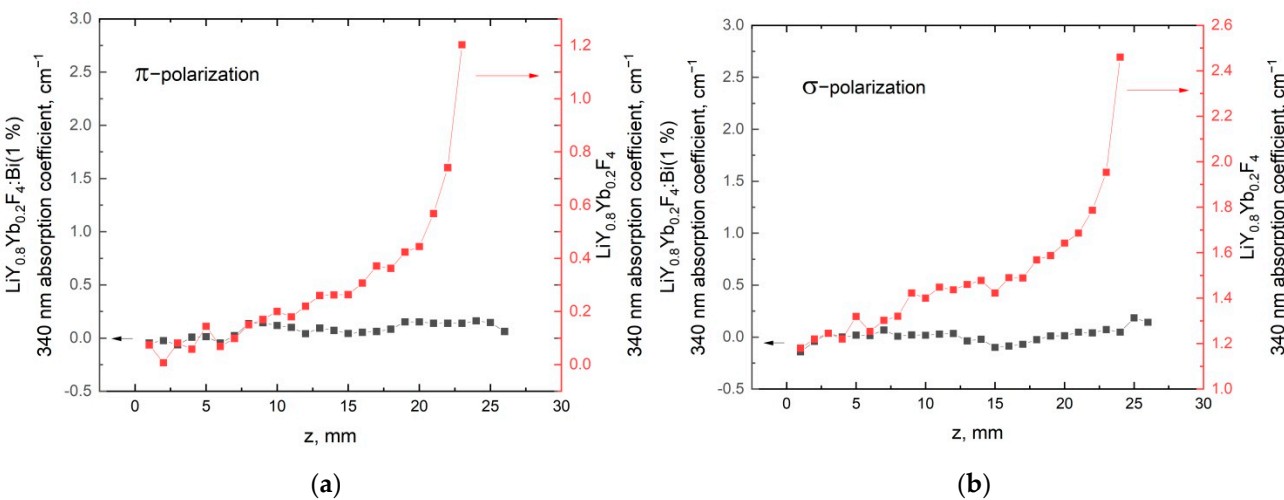

(**a**)                                                           (**b**)

**Figure 5.** Absorption coefficient at 340 nm along theLiY$_{0.8}$Yb$_{0.2}$F$_4$ crystals grown from the melt, with and without 1% of BiF$_3$ additive for $\pi$ (**a**) and $\sigma$ (**b**) polarizations.

The distribution coefficient of Yb$^{3+}$ ions in the LiYF$_4$ crystals should be close to 0.96, due to the isomorphic substitution of Y$^{3+}$ by the Yb$^{3+}$ ions [25]. In our LiY$_{0.8}$Yb$_{0.2}$F$_4$ sample, we observed the decrease in Yb$^{3+}$ concentration with a simultaneous increase in ultraviolet absorption (Figure 6) along the boule, from the beginning of crystallization to the end.

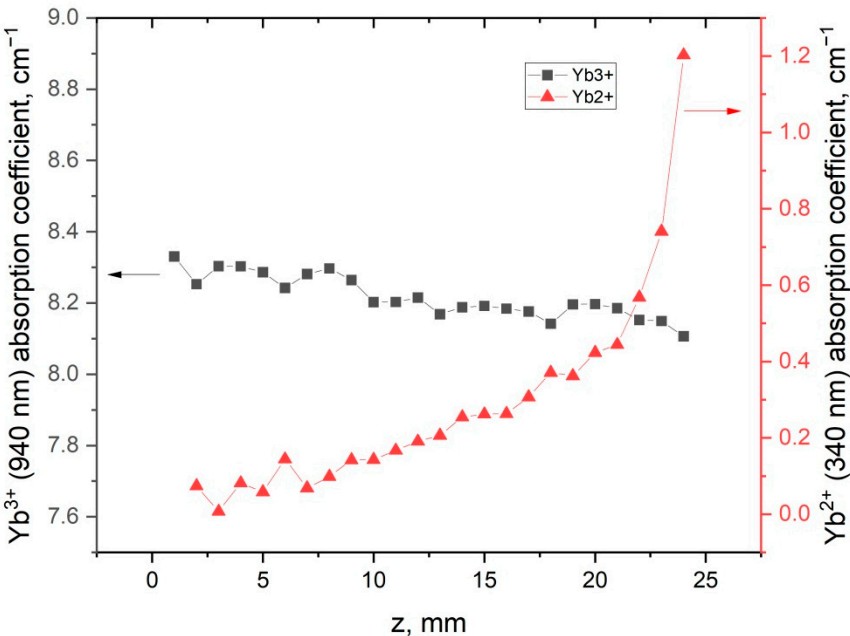

**Figure 6.** Yb$^{3+}$ and Yb$^{2+}$ absorption-coefficient along the LiY$_{0.8}$Yb$_{0.2}$F$_4$ crystal.

This relation between trivalent- and divalent-ytterbium absorption-coefficients is the indirect evidence of reduction processes occurring during crystal growth. Several studies devoted to the investigation of Yb$^{2+}$ distribution along fluoride crystals found that the segregation coefficient of Yb$^{2+}$ in YbF$_3$:CaF$_2$ crystals varies from 0.68 to 0.74, depending on Yb concentration [26]. In YbF$_3$:BaF$_2$, the segregation coefficient of divalent ytterbium equals 0.59 [24]. The implementation of formula (1) on the dependencies in Figure 6 gives the effective-distribution-coefficient values 0.98 and $-0.7$ for Yb$^{3+}$ and Yb$^{2+}$, respectively. Because $k_{eff}$ must be positive, the Yb$^{2+}$ ion content distribution along the crystal boule cannot be described by the Gulliver–Pfann equation. It can probably be explained by the high volatility of the bismuth salts and LiBiF$_4$ compound. It means that the chemical

composition of the melt and the crystallization conditions are drastically changed with the time of the crystal-growth process, which contradicts the basic assumptions in the derivation of the Galiver–Pfann law.

In addition, it has to be taken into account that, besides the significant rise in ultraviolet absorption, there is a drastic shortening of the luminescence-decay time of the $Yb^{3+}$ ions along the crystal without Bi doping, which is not evident for the $LiY_{0.8}Yb_{0.2}F_4$:$BiF_3$ (1%) sample (Figure 7).

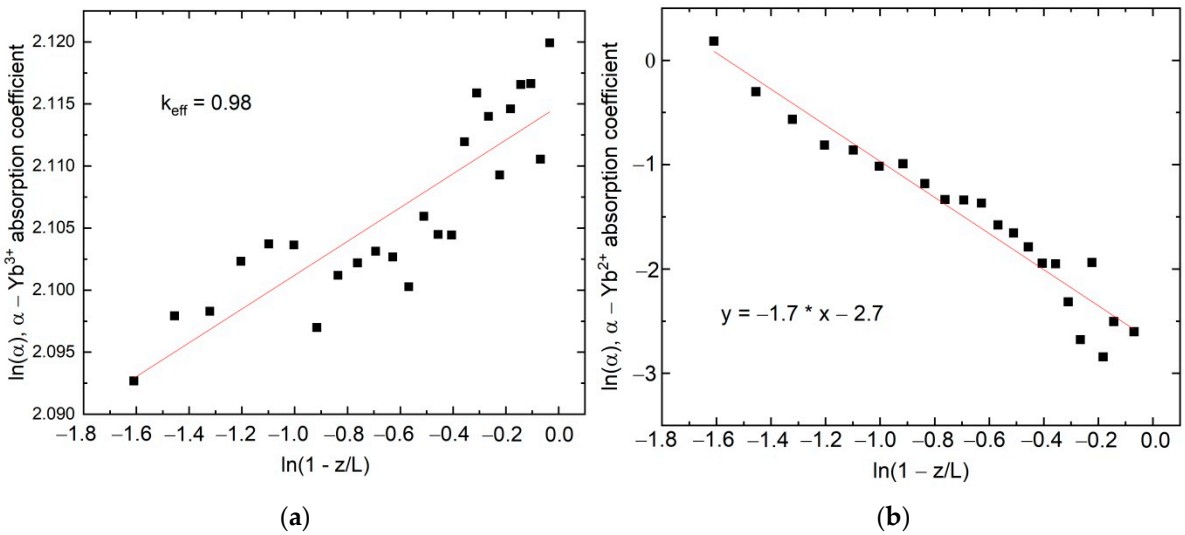

**Figure 7.** (**a**) $Yb^{3+}$ and (**b**) $Yb^{2+}$ absorption along the $LiY_{0.8}Yb_{0.2}F_4$ crystal in $\ln(\alpha)$ vs. $\ln(1-z/L)$ coordinates, where $\alpha$ is the absorption coefficient, z represents the beginning of the sample crystallization, and L is the full length of the sample.

The origin and the exact mechanism of the interaction require further investigation. It is known that $Yb^{3+}$ and $Yb^{2+}$ ions can form complex charge-transfer states [23]. It can be assumed that the $BiF_3$ additive sufficiently improves the optical quality of the $LiYF_4$:$Yb^{3+}$ materials, which were grown using the Bridgman technique in above-mentioned conditions. Finally, it was observed, that the addition of the $BiF_3$ additive to the charge sufficiently improves the optical quality of the $LiYF_4$:$Yb^{3+}$ materials, which were grown using the Bridgman technique in above-mentioned conditions.

It is also important to note that Bi doping cannot be the universal solution for the $Yb^{3+}$-$Yb^{2+}$ conversion problem, due to the near-infrared absorption of the low-valence Bi ions, therefore disturbing the original spectrum of the samples. In addition, the crystallization temperature must be higher than the volatilization temperature of the $BiF_3$. This fact imposes some limitations on the crystals.

## 4. Conclusions

For the first time, the distribution of $Yb^{2+}$ ions in $LiYF_4$:$Yb^{3+}$ crystals was investigated. It was suggested that the formation of $Yb^{2+}$ ions in the crystals occurs due to the growth conditions (for example, the presence of a trace quantity of water and/or synthesis in vacuum and graphite crucibles and heaters). The relation between $Yb^{2+}$ concentration and luminescence-decay time of $Yb^{3+}$ ions appeared to be clear and consistent. In particular, a significant increase in the $Yb^{2+}$ absorption coefficient occurs simultaneously with $Yb^{3+}$ luminescence quenching. The addition of $BiF_3$ to the melt led to the absence of the UV absorption bands corresponding to the $Yb^{2+}$ ions. In its turn, $Yb^{3+}$ luminescence-decay time remains stable for different parts of the crystals. The mechanism of energy transfer between $Yb^{2+}$ and $Yb^{3+}$ ions in heavily doped $LiYF_4$ crystals needs further investigation, because it can lower the effectiveness of the up-conversion excitation in co-doped $LiYF_4$:Yb:Tm/Ho/Eu laser crystals.

A positive effect of $BiF_3$ doping of the melt on the optical homogeneity of $LiYF_4$:$Yb^{3+}$ crystals has at least been detected.

We believe that one of the main advantages of the present work is that the work reveals some useful details concerning the $BiF_3$-based $LiYF_4$:$Yb^{3+}$ crystal growth. Indeed, the use of $BiF_3$ during the crystal-growth process seems to be beneficial compared to the alternative ways, such as the use of aggressive and toxic fluorinated-gases. In particular, in order to reduce the concentration of $Yb^{2+}$ in the i$YF_4$:$Yb^{3+}$ crystal, it is enough to add 17 mg of $BiF_3$ to the starting material to obtain 1 kg of $LiYF_4$:$Yb^{3+}$ crystals. In its turn, $BiF_3$ powder is relatively cheap, compared to fluorinated gases. The addition of $BiF_3$ powder allows the exclusion of several steps in the technology of the crystal-growth process. In particular, it allows for reducing the time of the drying of the starting material from a range of 12 to 20 h, to 2 to 4 h. In addition, the use of $BiF_3$ allows for the performance of the preparatory pumping of the vacuum chamber up to $10^{-1}$–$10^{-2}$ Pa, compared to $10^{-4}$–$10^{-5}$ Pa (without $BiF_3$). It reduces the time of crystal growth or lowers the requirements imposed on the vacuum-pump system.

**Author Contributions:** Conceptualization, A.K., S.K., O.M., A.N., V.S. and A.K.; formal analysis, M.P., M.G. and A.N.; investigation, A.K., N.A., S.K., O.M., A.N., V.S. and A.K.; resources, M.P., M.G. and A.N.; writing—original draft preparation, A.N., A.K.; writing—review and editing, A.N., V.S., M.P. All authors have read and agreed to the published version of the manuscript.

**Funding:** The work was funded by the subsidy allocated to Kazan Federal University for the state assignment in the sphere of scientific activities (project number FZSM-2022-0021).

**Data Availability Statement:** Not applicable.

**Conflicts of Interest:** The authors declare no conflict of interest.

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
