# Peer review of "The Impact of BiF3 Doping on the Yb3+ to Yb2+ Reduction during the LiYF4:Yb3+ Crystal-Growth Process"

_ceramics, doi:10.3390/ceramics5040085_

Round 1

Reviewer 1 Report

There are mainly the following problems:

1.      There was luminescence measuring instrument in Spectral-kinetic characterization of the samples, however, there was not any luminescence spectra in the manuscript. The change in luminescent of Yb3+ intensity before and after Bi ion doping is also important.

2.      The absorption of low valence Bi ion is very weak, and its absorption may not be measured. However, it will generate infrared luminescence under the excitation of near-infrared light, which will affect the infrared luminescence of LiY0.8Yb0.2F4 crystal. It is necessary to provide the luminescence spectra of the crystal before and after doping Bi under the excitation of 910-980nm light.

3.      The simplest and most sensitive way to test the existence of Yb2+ ion is to use ultraviolet lamp to irradiate and observe whether there is yellow light. Please measure the corresponding luminescent spectra.

4.      Some words in Figure 2 are too small to see clearly.

5.      What are the two vertical coordinates in Figure 3 (c)? What are their units?

Reviewer 2 Report

The article has some new results in the topic of Yb-doped LiYF4 crystals grown under 10-4 mbar vacuum. The authors proved the use of Bi-doping to suppress the presence of Yb2+ ions in the crystal. They also monitored the Yb2+ content by absorption measurements in the as grown LiYF4:Yb crystal and the energy transfer between Yb3+ and Yb2+ by lifetime measurements. Some remarks:

- There are typos in the text:

line 25: “YLF:Yb” instead of “LYF:Yb”

line 29: “mkm” instead of μm

line 45: “intevalence” needs to be “intervalence”

line 61: “104 mbar” instead of “10-4 mbar” 

line 24, 25, 55, 61 and 63 the indexes are in wrong places.

- One main remark, that the authors used a model to describe the concentration of dopants during the growth process which does not contain any time dependent source. A growth process in vacuum the reduction of Yb3+ to Yb2+ takes place in the meantime. They have to write some more comments in the text about this effect to explain the anomalous exponent value.

- In Figure 1. please clarify which parts of the crystals were measured?

- In Figure 3. the different plots cover each others!
